# Comparative Study of Bacterial Microbiota Differences in the Rumen and Feces of Xinjiang Brown and Holstein Cattle

**DOI:** 10.3390/ani14121748

**Published:** 2024-06-10

**Authors:** Haibo Lu, Shaokan Chen, Fengjie Li, Guoxing Zhang, Juan Geng, Menghua Zhang, Xixia Huang, Yachun Wang

**Affiliations:** 1Laboratory of Animal Genetics, Breeding and Reproduction, Ministry of Agriculture of China, National Engineering Laboratory of Animal Breeding, State Key Laboratory of Animal Biotech Breeding, College of Animal Science and Technology, China Agricultural University, Beijing 100193, China; luhaibo979@163.com (H.L.); bioin2@wekemo.com (G.Z.); 2Beijing Sunlon Livestock Development Company Limited, Beijing 100029, China; cowcode@163.com; 3College of Animal Science, Xinjiang Agricultural University, Urumqi 830052, China; fengjieli2021@126.com (F.L.); zhangmenghua810@126.com (M.Z.); 4Xinjiang Uygur Autonomous Region Animal Husbandry Station, Urumqi 830000, China; gengjuan124@163.com

**Keywords:** Xinjiang Brown, microbiota, Holstein, rumen, feces, breed

## Abstract

**Simple Summary:**

Xinjiang Brown cattle are a unique and widely distributed breed of dual-purpose cattle in Xinjiang, the People’s Republic of China. It is known that this cattle breed has different milk production performance compared to Holstein cattle. However, the microbiota differences between Xinjiang Brown and Holstein cattle are not well known. Therefore, this study aimed to compare the bacterial community composition of the rumen and feces of these two cattle breeds under the same dietary and management conditions using 16s rRNA sequencing. This study identified microbiota with different relative abundance between these two cattle breeds and their biological functions might be related to milk synthesis. Our study increases the understanding of the differences in bacterial microbiota between Xinjiang Brown and Holstein cattle.

**Abstract:**

Xinjiang Brown cattle are a unique and widely distributed breed of dual-purpose cattle in the Xinjiang area of China, whose milk production performance differs from Holstein cattle. It has been known that variations in bacterial species of the gastrointestinal tract influence milk protein, fat, and lactose synthesis. However, the microbiota differences between Xinjiang Brown and Holstein cattle are less known. This study aims to compare the bacterial community composition of the rumen and feces of these two cattle breeds under the same dietary and management conditions. The 16s rRNA sequencing data and milk production of 18 Xinjiang Brown cows and 20 Holstein cows on the same farm were obtained for analysis. The results confirmed differences in milk production between Xinjiang Brown and Holstein cattle. Microbiota with different relative abundance between these two cattle breeds were identified, and their biological functions might be related to milk synthesis. This study increases the understanding of the differences in microbiota between Xinjiang Brown and Holstein cattle and might provide helpful information for microbiota composition optimization of these dairy cattle.

## 1. Introduction

Xinjiang Brown cattle are one of the dual-purpose cattle breeds in the Xinjiang area, the People’s Republic of China [1], and account for a large proportion of local economic development for their superior local environmental adaptability [2,3]. Compared with specialized dairy cattle breeds like Holstein cattle, Xinjiang Brown cattle have a lower milk yield and higher milk fat and protein content [2,3], such that their average 305-day milk yield is 5469 kg, fat content is 4.13%, and protein content is 3.33%, respectively [1].

It is known that gastrointestinal microorganisms, e.g., bacteria, fungi, protists, and archaea, performing vital functions in plant fiber degradation and microbial cell protein synthesis, are important for milk synthesis [4,5,6]. Studies have reported that the structure of the bacterial community is associated with milk yield [7,8] and composition [7,9]. For example, the abundance of *Streptococcus*, *Ruminobacter*, and *Treponema* was higher in cows with a lower milk yield [4,10,11]. Therefore, we hypothesized that the phenotypical differences in milk production between Xinjiang Brown and Holstein cattle might partly result from their different microbial structure in the gastrointestinal tract.

It is suggested that Holstein cattle and buffalo carry different microbial species due to host genotype and microbiota interactions [12]. Different cattle breeds, e.g., Holstein and Jersey cows, also harbor different rumen bacterial communities [13]. However, a comparison of the microbiota composition of rumen and feces in the gastrointestinal tracts of Xinjiang Brown and Holstein cattle has not been studied yet.

This study aims to identify bacterial community composition in Xinjiang Brown and Holstein cattle under the same dietary conditions, using 16s rRNA sequencing. Understanding their differences in bacterial community composition might provide helpful information to optimize their bacterial community.

## 2. Materials and Methods

### 2.1. Animals and Sample Collection

The experimental animals, including 18 Xinjiang Brown (10 in the first parity and 8 in the second parity) and 20 Holstein (16 in the first parity and 4 in the second parity) cattle, were selected from the Urumqi State Seedstock Cattle Farm (Urumqi, China). All cows were fed ad libitum with the same total mixed ration diet for over 2 weeks. None of the cows were diagnosed with any diseases and had not been treated with any antibiotics for the last 3 months until sample collection. Sample collection occurred in the winter of 2018 (February). There were significant differences between the two breeds in terms of body condition score (Xinjiang Brown 3.81 ± 0.09 vs. Holstein 3.39 ± 0.09, *p*-value = 0.004 based on *t*-test) in similar days in milk (Xinjiang Brown 308.87 ± 21.37 vs. Holstein 305.00 ± 21.94, *p*-value = 0.551 based on *t*-test) at the sampling date.

Milk samples (30 mL) were collected in the morning, afternoon, and evening, followed by mixing in the proportion of 4:3:3. Preservatives were used to keep the milk samples fresh. Rumen liquid was collected using a flexible stomach tube, which was washed with clean water before each collection. About 50 mL of rumen liquid from each cow was aspired through the mouth, with the initial 100 mL (approximately) discarded to avoid contamination by saliva. Feces were sampled from the rectum at night (approximately 4 h after the night feed) and kept in 10 mL sterile centrifuge tubes. The obtained rumen fluid and feces samples were immediately placed into liquid nitrogen and were transferred to the laboratory for storage at −80°C. All the milk, rumen liquid, and feces samples were collected within 24 h.

### 2.2. Milk Production Performances Analysis

The daily milk yield (DMY) in a week (7 times) before sample collection was recorded. Milk composition, including milk fat content (Fat%), milk protein content (Protein%), milk lactose content (Lactose%), milk solids (Solids), somatic cell score (SCS), milk carbamide (Car), freezing point depression (FPD), and β-Hydroxybutyrate (BHB), was measured in the Xinjiang DHI centers in Urumqi based on component analysis models designed by Bentley Instruments Inc. (Chaska, MN, USA). This procedure was completed within 48 h after milk sample collection. All the above measurements were operated according to the manufacturer’s instructions.

### 2.3. DNA Extraction, PCR, and 16s rRNA Sequencing

Microbial DNA was extracted from the rumen liquid and feces samples of each animal using the DP328 DNA Kit (Tiangen Biotech, Beijing, China) according to standard protocols. The final DNA concentration and purification were determined by a NanoDrop 2000 UV–vis spectrophotometer (Thermo Scientific, Wilmington, NC, USA), and DNA quality was assessed by 1% agarose gel electrophoresis. The V3-V4 regions of the bacteria 16s rRNA gene were amplified with forward primers 338F 5′-ACTCCTACGGGAGGCAGCAG-3′ and reverse primer 806R 5′-GGACTACHVGGGTWTCTAAT-3′ by a thermocycler PCR system (GeneAmp 9700, ABI, Hampton, NH, USA). The PCR reactions were conducted using the following program: 3 min of denaturation at 95 °C, 27 cycles of 30 s at 95 °C, 30 s for annealing at 55 °C, 45 s for elongation at 72 °C, and a final extension at 72 °C for 10 min. PCR reactions were performed in triplicate in a 20 μL mixture containing 4 μL of 5 × FastPfu Buffer, 2 μL of 2.5 mM dNTPs, 0.8 μL of each primer (5 μM), 0.4 μL of FastPfu Polymerase, and 10 ng of template DNA. The PCR products were extracted from 2% agarose gel, purified using the AxyPrep DNA Gel Extraction Kit (Axygen Biosciences, Union City, CA, USA), and quantified using QuantiFluor™-ST (Promega, Madison, WI, USA) according to standard protocol. The amplicon library was paired-end sequenced (2 × 300 bp) on an Illumina MiSeq platform (Illumina, San Diego, CA, USA) according to standard protocol.

### 2.4. Sequence Data Processing

All the bioinformatic analysis procedures of the sequence data were completed on the Quantitative Insights Into Microbial Ecology (QIIME2) platform [14]. The sub-operational taxonomic unit (sOTU) is an OTU at a similarity of 100%, and, thus, 2 sequences from the same region of 16s rRNA with a single-nucleotide difference belong to two different sOTUs. Deblur integrated into QIIME2 was used to obtain putative error-free sequences and pick sOTU [15]. The representative sequence of each sOTU was then aligned to the Silva database Release 132 [16] to assign taxonomy using the q2-feature-classifier [17]. Any contaminating mitochondrial and chloroplast sequences were filtered out using the QIIME2 feature-table plugin. This resulted in 83% of the total sequences being assigned taxonomically at the genus level. The relative abundance table of bacteria was calculated for the following bacterial statistical analysis. Unless specified above, the parameters used in the analysis were set to defaults.

### 2.5. Bioinformatic and Statistical Analysis

An SAS general linear model (GLM) procedure [18] considering the fixed effects of the breed was used to evaluate the differences in milk production performances between breeds. Principal coordinates analysis (PCoA) of Bray–Curtis dissimilarity [19], which was calculated from the sOTU sequence count table, was used to visualize the differences in the bacterial community among samples and between breeds, followed by Permutational Multivariate Analysis of Variance (PERMANOVA), a non-parametric multivariate statistical test [20], to test the significance of the differences in the bacterial community between breeds and the site of the gastrointestinal tract.

Microbial biomarkers, including bacterial taxa from the phylum to genus level, were identified using the Linear discriminant analysis Effect Size (LEfSe) analysis in the MicrobiotaProcess package in R [21]. The threshold values of the logarithmic LDA score were set to 2.

## 3. Results and Discussion

### 3.1. Differences of Milk Production Performances between Xinjiang Brown and Holstein Cattle

The SAS GLM analyses of milk production performance between Xinjiang Brown and Holstein cattle are shown in Table 1. The daily milk yield in Xinjiang Brown cattle was about 8 kg lower than Holstein cattle. However, the milk fat content, protein content, and milk solids were higher in Xinjiang Brown cattle than in Holstein cattle. Milk carbamide and freezing point depression were higher in Xinjiang Brown cattle than in Holstein cattle. In addition, the milk lactose content, SCS, and BHB were not different between these 2 cattle breeds. The differences in milk production performance between Xinjiang Brown and Holstein cattle were consistent with the literature [2,3], and these differences in milk performance indicated physiological diversity in these two breeds.

### 3.2. Microbiota Diversity between Xinjiang Brown and Holstein Cattle

As DNA extraction from one Holstein feces sample failed, 3,645,770 available raw reads were obtained from 38 rumen and 37 feces samples with average reads of 48,610 ± 6813 per sample. After deleting the chimeric reads, the average number of reads was 23,055 ± 4536 per sample. The differences between microbial communities in the rumen and feces samples of these two breeds were examined by PCoA based on Bray–Curtis dissimilarity (Figure 1). The bacterial community profiles in the rumen and feces were distinct from each other, with the rumen samples exhibiting higher variance than the feces samples (Figure 1A, R^2^ = 0.531, *p* < 0.001). In addition, the breed displayed a significant effect on bacterial community in the rumen (Figure 1B, R^2^ = 0.098, *p* < 0.001) and feces (Figure 1C, R^2^ = 0.071, *p* < 0.001). Note that the dissimilarity between rumen and feces was larger than the dissimilarity within rumen/feces of different breeds.

The microbiota composition along the length of the gastrointestinal tract generally clusters into different groups [22,23,24], which was also confirmed in our research. This is because the microbiota community in the rumen has different functions for cows’ digestive ability compared with the microbes in the rectum [25,26,27]. Studies demonstrate that quantitative trait loci are linked to microbial taxa in the gastrointestinal tract [28,29], and host genotype can affect microbial community composition [30,31]. Our results showed that, to a larger extent, the microbial communities of Xinjiang Brown and Holstein cows were similar in the rumen and feces, respectively. This is consistent with studies comparing the similarity in the microbial composition between evolutionarily distant hosts [26]. Therefore, despite the production and physiological divergence between these two cattle breeds, the function of the rumen and rectum is the main factor affecting bacterial community composition.

### 3.3. Relative Abundance of Microbiota at the Phylum Level between Xinjiang Brown and Holstein Cattle

The microbial communities in different breeds at the phylum level are shown in Figure 2. The main phyla in rumen samples were *Bacteroidetes*, followed by *Firmicutes*, *Proteobacteria*, *Spirochaetes*, and *Patescibacteria*. By contrast, the main phyla in feces samples were *Firmicutes*, followed by *Bacteroidetes*, *Spirochaetes*, and *Tenericutes*. The relative abundance of bacterial phyla is shown in Appendix A. Compared to Holstein cattle, the Xinjiang Brown cattle showed a decrease in the proportions of *Spirochaetes* in rumen samples and a decrease in the proportions of *Bacteroidetes* in feces samples, and no other changes in microbiota at the phylum level were identified. The differences in bacterial community between the rumen and feces samples were larger than differences within rumen/feces samples of different breeds, which is consistent with the results in Figure 1.

In line with most studies, the bacterial communities of both the rumen and manure samples were dominated by the phyla *Bacteroidetes* and *Firmicutes* [32,33]. Compared with the rumen samples, the feces samples contained a greater abundance of *Firmicutes* and a lower abundance of *Bacteroidetes* [33]. The changing ratios between the two major phyla might be related to the different bacterial functions passing through the ruminant’s digestive tract [32].

### 3.4. Significant Differences in Microbiota Composition between Holstein and Xinjiang Brown Cattle

In total, 19 bacterial phyla, 29 classes, 47 orders, 84 families, and 216 genera were identified in rumen samples; 12 bacterial phyla, 19 classes, 31 orders, 67 families, and 184 genera were identified in feces samples. The bacteria with more than 0.1 % relative abundance were used for further analyses. The bacterial taxa with significant differential relative abundance were identified at the phylum, class, order, family, and genus level between rumen and feces samples in different breeds based on LEfSe analyses (Figure 3). For the rumen sample, 18 clades significantly differed between Xinjiang Brown and Holstein cattle. There were higher (*p* < 0.05 and log_10_(LDA Score) > 2.0) proportions of the genera *Ruminococcaceae_NK4A214_group*, *Succiniclasticum*, *Succinivibrio*, family *Acidaminococcaceae*, order *Selenomonadales*, and class *Negativicutes* in Xinjiang Brown cattle than in Holstein cattle, whereas the relative abundance of 12 clades was higher (*p* < 0.05 and log_10_(LDA Score) > 2.0) in Holstein cattle compared to Xinjiang Brown cattle, including genera *Butyrivibrio_2*, *Lachnospiraceae_NK4A136_group*, *Pseudobutyrivibrio*, *Ruminococcus_1*, and *Anaerovibrio*; family *Spirochaetaceae*; orders *Absconditabacteriales_(SR1)*, *Spirochaetales*, and *Mollicutes_RF39*; classes *Gracilibacteria* and *Spirochaetia*; and phylum *Spirochaetes*. The *Ruminococcaceae_NK4A214_group* and *Ruminococcus_1* were both from the *Ruminococcaceae* family, with *Ruminococcaceae_NK4A214_group* being higher in Xinjiang Brown cattle and *Ruminococcus_1* being higher in Holstein cattle. The genera *Succiniclasticum* and *Anaerovibrio* were from the same order, with *Succiniclasticum* being higher in Xinjiang Brown cattle and *Anaerovibrio* being higher in Holstein cattle. The *Butyrivibrio_2*, *Lachnospiraceae_NK4A136_group* and *Pseudobutyrivibrio* in the *Lachnospiraceae* family were both lower in Xinjiang Brown cattle than in Holstein cattle.

For the feces samples, 28 clades significantly differed between Xinjiang Brown and Holstein cattle. There were higher (*p* < 0.05 and log_10_(LDA Score) > 2.0) proportions of the genera *Dorea*, *Lachnospiraceae_UCG-010*, *Romboutsia*, *Flavonifractor*, *Ruminococcaceae_UCG-004*, *Ruminococcaceae_UCG-005*, *Ruminococcus_2*, *Turicibacter*, and *Anaerovibrio*; families *Lachnospiraceae*, *Peptococcaceae*, *Peptostreptococcaceae*, *Veillonellaceae*, and *p-2534-18B5 gut group*; and order *Mollicutes_RF39* in Xinjiang Brown cattle than in Holstein cattle, whereas 13 clades were higher (*p* < 0.05 and log_10_(LDA Score) > 2.0) in relative abundance in Holstein cattle compared to Xinjiang Brown cattle, including the genera *Clostridium_sensu_stricto_1*, *Anaerosporobacter*, *Coprococcus_3*, *Tyzzerella_4*, *[Eubacterium]_coprostanoligenes_group*, *Ruminococcaceae_UCG-009*, and *Succinivibrio*; families *Clostridiaceae_1*, *Succinivibrionaceae*, and *Bacteroidales_RF16_group*; order *Aeromonadales*; class *Gammaproteobacteria*; and phylum *Proteobacteria*.

Our results showed that *Prevotellaceae* was the dominant family in the rumen sample, which is consistent with the literature [34]. The genus *Prevotella* utilizes starch and proteins to produce succinate and acetate. It has been reported that total milk solid content is positively correlated with the abundances of *Prevotellaceae UCG-001* and *Prevotellaceae UCG-003* [4]. However, in our study, no genus in *Prevotellaceae* showed different relative abundance between Xinjiang Brown and Holstein cattle.

*Succiniclasticum* had a higher relative abundance in the rumen samples of Xinjiang Brown cattle (5.5%) than Holstein cattle (5.3%) (Figure 3). *Succiniclasticum* is affiliated with the order *Selenomonadales* and specializes in fermenting succinate and converting it to propionate [35,36]. As the primary succinate-utilizing bacteria, a higher level of *Succiniclasticum* has been associated with a greater production of succinate from starch degradation [37,38]. Xue et al. reported that *Succiniclasticum* is higher in Holstein cows with a high milk yield and high milk protein content than in cows with a low milk yield and low milk protein content [39]. Xinjiang Brown cattle had a higher relative abundance of *Succiniclasticum*, higher milk solids, higher milk fat yield, and lower daily milk yield compared with Holstein cattle; these results are consistent with the literature [7,40] that states that *Succiniclasticum* is negatively correlated with milk yield and positively correlated with milk fat yield.

In the rumen samples, the *Ruminococcaceae_NK4A214_group* was higher in Xinjiang Brown cattle, and *Ruminococcus_1* was higher in Holstein cattle. In the feces samples, *Ruminococcaceae_UCG_004*, *Ruminococcaceae_UCG_005*, *Ruminococcus_2*, and *Flavonifractor* were higher in Xinjiang Brown cattle, and *Ruminococcaceae_UCG_009* and *[Eubacterium]_coprostanoligenes_group* were higher in Holstein cattle (Figure 3). They are all in the *Ruminococcaceae* family and are considered vital for rumen fiber degradation and cellulose decomposition [32], breaking down fibrous plant materials to produce acetate, formate, and succinate [33,39,41]. Liu et al. revealed that total milk solid content was positively correlated with the abundances of the *Ruminococcaceae NK4A214 group*, *Ruminococcaceae UCG_014*, and *Ruminococcus_2* in Holstein cattle rumen samples [4]. The genera *RuminococcaceaeNK4A214_group*, *Ruminococcus_2*, and *[Eubacterium]_coprostanoligenes_group* were more abundant in high-yield lactating Holstein cows [38]. Similar to *Ruminococcaceae*, the *Lachnospiraceae* family also plays predominant roles in biohydrogenation pathways [38,42]. In animals fed a high-fiber diet, fibrolytic and cellulolytic bacteria, *Ruminococcaceae* and *Fibrobacteraceae*, were found to have the highest abundance compared with animals fed a lower-fiber diet [43]. The literature revealed that total milk solid content was positively correlated with the abundances of the *Lachnospiraceae_XPB1014_group* [4]. The genus *Lachnospiraceae_BS11_gut_group* was more abundant in the high-yield lactating Holstein cows [38]. Our results showed that in the rumen sample, *Butyrivibrio_2*, *Lachnospiraceae_NK4A136_group*, and *Pseudobutyrivibrio* in the *Lachnospiraceae* family were lower in Xinjiang Brown cattle than in Holstein cattle. In the feces samples, *Dorea* and *Lachnospiraceae_UCG-010* were higher in Xinjiang Brown cattle, whereas *Anaerosporobacter*, *Coprococcus_3*, and *Tyzzerella_4* were higher in Holstein cattle (Figure 3). Zhang et al. found that *Coprococcus 3* was more abundant in Holstein cow feces and strongly and positively related to milk yield [24], which is consistent with our findings, even though the feed might differ from our experiment. The results showed that different domain genera from the *Ruminococcaceae* and *Lachnospiraceae* families are abundant in Xinjiang Brown and Holstein cattle, and this biological background needs further research.

*Anaerovibrio* were higher in Xinjiang Brown feces samples than in Holstein samples, with a low relative abundance in both breeds (Figure 3). Xue et al. found that the relative abundances of *Anaerovibrio* were not different between cows with high milk yield and high milk protein content and cows with low milk yield and low milk protein content [39]. The genus *Butyrivibrio* has functions that include fiber degradation, protein breakdown, and butyrate production. Milk fat was positively correlated with the relative abundances of *Butyrivibrio* [34], and total milk solid content was positively correlated with the abundance of *Butyrivibrio 2* [4]. However, *Butyrivibrio* was not different between Xinjiang Brown and Holstein breeds. These inconsistencies in the results with the literature might be due to the differences in cattle breeds and need confirmation in further research.

## 4. Conclusions

This study explores the taxonomic diversity in microbial communities in the rumen and feces of Xinjiang Brown and Holstein cows through a metagenomic analysis. The results showed that the breed’s effect on microbiota composition was smaller than that of the sample site. Bacterial taxa with significant differential relative abundance in rumen and feces samples were identified between these two cattle breeds, and their biological functions might be related to milk synthesis. This study contributes to the understanding of microbiota differences between Xinjiang Brown and Holstein cattle and might provide helpful information for unraveling the relationships between host milk production performance and gut microbiota.

## Figures and Tables

**Figure 1 animals-14-01748-f001:**
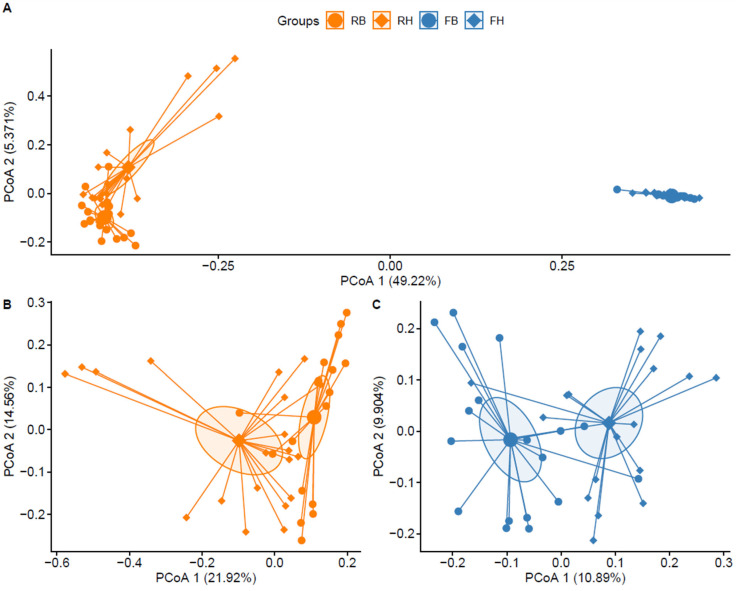
PCoA of the microbiota in rumen samples in Xinjiang Brown cattle (RB), rumen samples in Holstein cattle (RH), feces samples in Xinjiang Brown cattle (FH), and feces samples in Holstein cattle (FH). (**A**) Bray–Curtis PCoA based on the relative abundance of the OTU of all samples. (**B**,**C**) Bray–Curtis PCoA based on the relative abundance of the OTU of the rumen or feces samples. The different points represent different samples. The distance between the points shows the degree of discrepancy in the microbial structure of the samples.

**Figure 2 animals-14-01748-f002:**
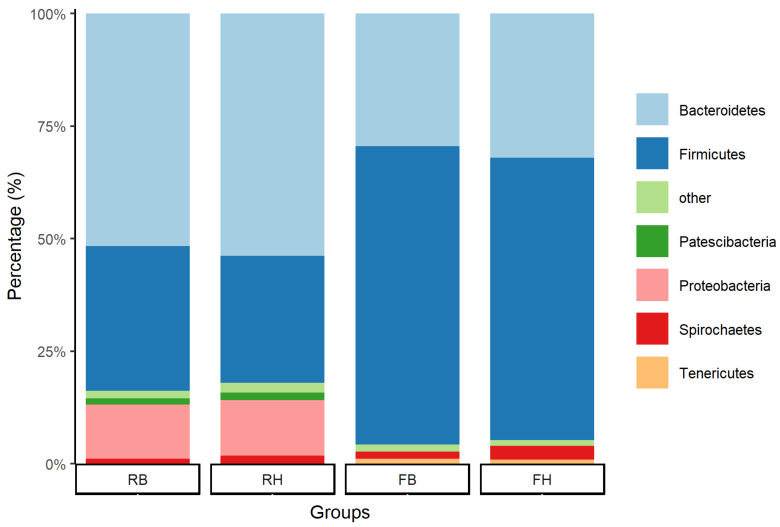
The relative abundance of bacterial phyla in the rumen of Xinjiang Brown (RB) and Holstein cattle (RH) and the feces of Xinjiang Brown (FB) and Holstein cattle (FH).

**Figure 3 animals-14-01748-f003:**
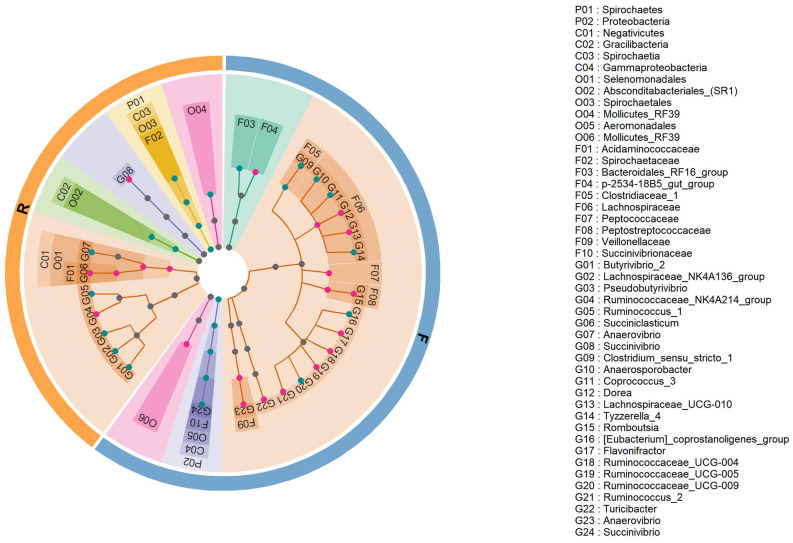
Hierarchical tree based on differential microbial composition at the level of the phylum (P), class (C), order (O), family (F), and genus (G) between rumen (left section in orange) and feces (right section in blue) samples. Each node (small circle) represents a taxon; red nodes represent bacterial biomarkers with higher relative abundance in Xinjiang Brown cattle than in Holstein cattle, whereas green nodes represent bacterial biomarkers with lower relative abundance in Xinjiang Brown cattle than in Holstein cattle.

**Table 1 animals-14-01748-t001:** Differences in milk production performances between Xinjiang Brown and Holstein cattle.

Item	Least Squares Mean ± SE		*p*-Value
	Xinjiang Brown	Holstein	
DMY	18.71 ± 1.22	24.68 ± 1.25	0.002
Fat	4.94 ± 0.16	3.66 ± 0.16	<0.001
Protein	3.91 ± 0.07	3.49 ± 0.07	<0.001
Solids	14.40 ± 0.16	12.79 ± 0.17	<0.001
Car	19.29 ± 0.70	11.81 ± 0.71	<0.001
FPD	540.55 ± 1.90	524.46 ± 1.95	<0.001
SCS	820.00 ± 206.35	451.42 ± 211.82	0.217
Lactose	5.03 ± 0.03	5.03 ± 0.03	0.977
BHB	0.87 ± 0.09	0.87 ± 0.09	0.979

## Data Availability

The data that support the findings of this study are available from the corresponding author, Yachun Wang, upon reasonable request.

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
