# Peer review of "Comparative Study of Bacterial Microbiota Differences in the Rumen and Feces of Xinjiang Brown and Holstein Cattle"

_animals, 2024, doi:10.3390/ani14121748_

Round 1

Reviewer 1 Report

Comments and Suggestions for Authors

The manuscript describes interesting findings, principally for the chinese researchs and produces. However, there are significant points that needs to be clarify by the authors in order to be published. Most of my considerations you will find on the original PDF file.

Comments on the Quality of English Language

About of quality of English language, some sentences needs to be revised in order to make clear for the reader.

Author Response

Thanks for your time and constructive comments. Your comments helped us to improve the manuscript.

For the response, please see the attachment.

Reviewer 2 Report

Comments and Suggestions for Authors

The manuscript “Comparative study of the rumen and feces microbiota differences in Xinjiang Brown and Chinese Holstein cattle” describes results of an original study performed using the contemporary and relevant method of NG, and contains new data that may be interesting for specialists in cattle microbiota, breeding and milk manufacturing. This manuscript may be published after major and minor corrections.

Major corrections

Lines 19-20, 32-34, 292-293. The authors claim that “This study identified the bacterial with different relative abundance between these 2 cattle breeds and their biological function mainly related to milk synthesis.” Such statement is absolutely incorrect.

Unfortunately, milk synthesis is not a function of microbiota at all. This process is a result of the complex interaction between cattle metabolism and heredity, gastrointestinal microbiota and environmental factors. The authors only have found some bacterial taxa with significant differential abundance between two cattle breeds, and described correlations of the taxa with milk production performance based on the literature data. If the authors would like to get direct relationship between abundance of the bacterial taxa and performance of milk production, they should perform numerous experiments with selective enrichment or deletion of those taxa in cattle individuals in tens and hundreds replicates. Without such experiments the authors may at least suggest that the revealed differences of bacterial taxa with significant differential abundance between two cattle breeds may be related to the differences in performance of milk production between the breeds, but not more.

For this reason, all the statements mentioned above on “biological function of bacteria mainly related to milk synthesis” should be removed in the text of the manuscript and exchanged by real results and relevant conclusions.

Lines 59-61. The authors claim that “However, the comparison of rumen and feces microbiota composition in the gastrointestinal tract for Xinjiang Brown and Chinese Holstein cattle remain less characterized.” But, the examples of such studies are not cited. The authors should either directly indicate that microbiota of the cattle breeds has not been studied yet, or cite respective sources.

Minor corrections

Line 19. “…bacterial…” should be changed by “…bacteria…”.

Line 49. “…micro-organism…” should be changed by “…microorganisms…”.

Line 50. “…degraders…” should be changed by “…degradation…”.

Line 144. “…were…” should be changed by “…was…”.

Line 145-146. “…average reads…were obtained.” should be changed by “average number of reads…was obtained.”.

Lines 169-171. The phrase “Our results showed that to a larger extent the predominant microbial community of bacterial of Xinjiang brown and Holstein cows were similar in the rumen and feces communities, respectively.” is incorrect and complicated. It is recommended to modify it like this “Our results showed that to a larger extent the microbial communities of Xinjiang brown and Holstein cows were similar in the rumen and feces, respectively.”

Line 175. “…were showed…” should be changed by “…are shown…”.

Line 180. “The relative abundance of bacterial at phylum level were shown…” should be changed by “The relative abundances of bacterial phyla are shown …”.

Lines 189-191. The legend to Figure 2 is too long and complicated. It is recommended to modify it like this “Bar chart of bacterial phyla relative abundance in rumen and feces samples of Xinjiang Brown (RB) and Holstein (RH) cattle.”

Lines 198, 199 and elsewhere. “Significant differences… or …significant bacterial differences…” is incorrect term. The correct one is “bacterial taxa with significant differential abundance”.

Lines 202, 206, 224, 229. “Genus” in plural is “genera”.

Lines 207-208, 222-231. The authors should use plural names of “family”, “order” and “class”, namely “families”, “orders” and “classes”.

Comments on the Quality of English Language

Numerous grammar errors including wrong verb forms, singular/plural as well as improper, incorrect and complicated phrases should be corrected.

Author Response

(The authors gave the same response as above.)

Round 2

Reviewer 1 Report

Comments and Suggestions for Authors

In my understanding most of the suggestions have been accepted, and the manuscript has improved. So my recommendation is to published.

Author Response

Thank you again for your effort and constructive comments.

Reviewer 2 Report

Comments and Suggestions for Authors

The manuscript “Comparative study of the rumen and feces microbiota differences in Xinjiang Brown and Chinese Holstein cattle” describes results of an original study performed using the contemporary and relevant method of NG, and contains new data that may be interesting for specialists in cattle microbiota, breeding and milk manufacturing. This manuscript has been improved by the authors, and may be published after minor corrections.

Line 33. “…bacterial…” should be changed by “…bacteria…”.

Line 52. “…microorganism…” should be changed by “…microorganisms…”.

Comments on the Quality of English Language

Proofreading is necessary, because the numerous grammar errors are present in the text. For instance, the authors use passive voice with verbs in the second form instead of the third form. For instance, in Line 214 it is erroneous "are showed" instead of "are shown".

Author Response

Authors’ response to the reviewer 2:

Thanks for your time and constructive comments. Your comments helped us to improve the manuscript.

Comments to the Authors:

The manuscript “Comparative study of the rumen and feces microbiota differences in Xinjiang Brown and Chinese Holstein cattle” describes results of an original study performed using the contemporary and relevant method of NG, and contains new data that may be interesting for specialists in cattle microbiota, breeding and milk manufacturing. This manuscript has been improved by the authors, and may be published after minor corrections.

Line 33. “…bacterial…” should be changed by “…bacteria…”.

Au: We changed the manuscript (Line 32). 

Line 52. “…microorganism…” should be changed by “…microorganisms…”.

Au: We changed the manuscript (Line 46). 

Proofreading is necessary, because the numerous grammar errors are present in the text. For instance, the authors use passive voice with verbs in the second form instead of the third form. For instance, in Line 214 it is erroneous "are showed" instead of "are shown".

Au: We changed the manuscript (Lines 141, 185, 189). We also checked the grammar and words throughout the manuscript and highlighted the main changes.